# Green Route to Produce Silver Nanoparticles Using the Bioactive Flavonoid Quercetin as a Reducing Agent and Food Anti-Caking Agents as Stabilizers

**DOI:** 10.3390/nano12193545

**Published:** 2022-10-10

**Authors:** Sofia L. Ramírez-Rosas, Enrique Delgado-Alvarado, Luis O. Sanchez-Vargas, Agustin L. Herrera-May, Mariana G. Peña-Juarez, J. Amir. Gonzalez-Calderon

**Affiliations:** 1Faculty of Science, Universidad Autónoma de San Luis Potosí, San Luis Potosí 78290, Mexico; 2Micro and Nanotechnology Research Center, Universidad Veracruzana, Boca del Rio 94294, Mexico; 3Oral Biochemistry and Microbiology Laboratory, Faculty of Stomatology, Universidad Autónoma de San Luis Potosí, Av. Manuel Nava No. 64, San Luis Potosí 78290, Mexico; 4Master in Applied Engineering, Faculty of Contruction Engineering and Hábitat, Universidad Veracruzana, Boca del Rio 94294, Mexico; 5Institutional Doctorate in Engineering and Materials Science, Universidad Autónoma de San Luis Potosí, Sierra Leona No. 550 col. Lomas 2da. Sección, San Luis Potosí 78210, Mexico; 6Cátedras Conacyt- Institute of Physics, Universidad Autónoma de San Luis Potosí, Álvaro Obregón 64, San Luis Potosí 78000, Mexico

**Keywords:** AgNPs, nanostructure, X-ray Diffraction (XRD), environmentally protective

## Abstract

In previous work, the isolated polyphenolic compound (PPC) quercetin was used as a reducing agent in the formation of silver nanoparticles (AgNPs), testing two types of quercetin. This PPC is a bioactive molecule that provides the electrons for the reduction of silver ions to zerovalent silver. The results demonstrated that quercetin in dietary supplement presentation was better than reagent grade quercetin for the synthesis of AgNPs, and the difference between them was that the dietary supplement had microcrystalline cellulose (CM) in its formulation. Therefore, this dietary anti-caking agent was added to the reagent-grade quercetin to validate this previously found improvement. AgNPs were obtained at neutral pH by a green route using quercetin as a reducing agent and microcrystalline cellulose and maltodextrin as stabilizing agents. In addition, different ratios were evaluated to find the optimum ratio. Ultraviolet-Visible spectroscopy (UV-VIS), Atomic Force Microscope (AFM), Z-potential, Dynamic Light Scattering (DLS) and X-ray Powder Diffraction (XRD) were used for characterization. The antibacterial activity of the *S. aureus* and *E. coli* agent was tested by the disk diffusion and microdilution method. According to the results, this green synthesis needs the use of food stabilizer when working at pH 7 to maintain AgNPs in the long term. The ideal ratio of reducing the agent:stabilizing agent was 1:2, since with this system stable AgNPs are obtained for 2 months and with improved antimicrobial activity, validating this method was ecologically and economically viable.

## 1. Introduction

For centuries, inorganic antimicrobial agents have been used to combat pathogenic bacteria. For example, silver (Ag) has been used to combat infection and control microbial contamination for centuries. The main application of antibacterial nanoparticles is in the medical field for the development of dressing, coating, and drug delivery, since they are an alternative to the growing resistance of microorganisms to antibiotics [1]. Recently, silver nanoparticles (AgNPs) have demonstrated greater antimicrobial activity than bulk silver. The above, due to their antimicrobial capacity, is due to mechanisms such as the perturbation of cell membrane functions (altering permeability and cellular respiration), entrance of nanoparticles into the cell (perturbing protein and deoxyribonucleic acid (DNA) function), or production of oxidative species due to the presence of particles inside the cell [2].

However, conventional methods used for the formation of AgNPs usually involve complicated reactions and materials, including compounds such as sodium borohydride, hydrazine, and hypophosphite. These materials are ecologically incompatible, since they are toxic compounds that are often adsorbed on the surface of the nanoparticle and are not likely to be eliminated during the synthesis process. They may also require high temperatures, alkaline pH (with values of 9 to 11), and vacuum-assisted procedures; in addition, because of their high reactivity caused by their large surface area, special care is needed for their washing and filtering to avoid the hydrolysis and unwanted oxidation of the materials [3]. All of the above is in addition to the contamination or management of resources for the proper handling of hazardous waste.

Faced with these drawbacks is that “green” synthesis emerged, with the help of environmentally friendly technology, which is based on the development of cleaner and low-temperature methods, and on the use of natural extracts, especially those containing polyphenolic compounds (PPCs). The PPCs possess aromatic benzene rings substituted by hydroxyl (-OH) groups, which participate in the reduction of free metal ions, i.e., silver ions (Ag^−^) to silver nanoparticles (AgNPs). Therefore, natural extracts, and foods such as purple onion, teas, and apples, have been used to synthesize AgNPs. However, this technology is limited because natural extracts come from fruits and roots that vary widely in their content of bioactive molecules due to the moisture content, ripening stage, and other compounds in the food [4,5]. 

Quercetin, a main PPC in nature, is found in the roots, leaves, and fruits of plants. Therefore, these natural materials have been used to reduce ions to nanoparticles, thanks to their high content of quercetin. Previous studies [6,7] have compared the performance of natural extracts with reagent-grade quercetin, which was 99% pure, and found that the performance of both was similar; nevertheless, it was necessary to modify the pH to 10 when reagent-grade quercetin was used. Otherwise, the size, quantity, and stability of the nanoparticles were not suitable to be used, since they became agglomerated and lost their nanometric size. In our previous works [7,8], we proved that using quercetin in a food supplement presentation with a quercetin content of 20% and with microcrystalline cellulose (CM) managed to obtain AgNPs of better size, shape, and long-term stability than when reagent-grade quercetin was used, and without the need to change the quercetin solution at alkaline pH. From the above, it was proposed that the presence of CM was responsible for maintaining the nanometer size of AgNPs for a prolonged period, as it serves as a stabilizing agent. CM is an emulsifier and anti-caking in the food industry.

CM is a derivative of depolymerized α-cellulose, in which the cell wall of the fiber has been broken down into fragments ranging in size from tens to a few hundred microns. CM is a fine white powder, odorless and inert. On an industrial scale, microcrystalline cellulose is obtained by the hydrolysis of wood and cellulose cotton. It can also be obtained from coconut husks, sugar cane bagasse, ramie, wheat straw, rice, jute, flax, and linseed. It is used in food production as an anti-caking agent and emulsifier. Commercially, it is found in vitamin supplements and tablets [9,10]. Another similar stabilizing agent in the food industry is maltodextrin (MD), which is a mixture of several glucose polymers resulting from the partial hydrolysis of starch, usually from corn, rice, potato, or wheat starch. MD is classified according to its dextrose equivalence and is commercially presented as a fine white powder with a neutral flavor. It is usually used in hard-drying materials as an anti-caking agent and stabilizer [11].

Having these studies as a background and according to the information gathered, it was decided to carry out the experimentation to obtain AgNPs through a green synthesis using the flavonoid quercetin at pH 7 as a reducing agent and to prove that the use of emulsifying and anti-caking agents of natural origin, commonly used in the food industry, CM and MD, stabilized the AgNPs for a long time for their intended use. Therefore, pure quercetin was used as a reducing agent to obtain and to improve the stability of the AgNPs, and CM and MD were used as stabilizing agents. Different ratios (by weight) with respect to the amount of quercetin were tested with the aim of finding the optimal ratio that enhances the formation, long-term stabilization and antimicrobial properties of these AgNPs against Gram-positive and Gram-negative bacteria.

## 2. Materials and Methods

### 2.1. Materials 

Silver Nitrate (AgNO_3_, 99.8%) obtained from Productos Químicos Monterrey Fermont (Monterrey, Mexico) was used. Quercetin reagent grade (QR) (>95%) was supplied by Sigma Aldrich (New Delhi, India). Microcrystalline Cellulose 102 and Maltodextrin D10 from Encapsuladoras Mexico (Ciudad de Mexico, Mexico) were also used.

#### 2.1.1. Synthesis of Silver Nanoparticles (AgNPs) at a Different Ratio of Stabilizing Agent

A total of 11 systems were prepared, varying the ratio of the stabilizing agents in relation to pure quercetin QR (1:0). For microcrystalline cellulose: QR: CM (1:0.5), QR: CM (1:1), QR: CM (1:2), QR: CM (1:3), and QR: CM (1:4), and for maltodextrin: QR: MD (1:0.5), QR: MD (1:1), QR: MD (1:2), QR: MD (1:3), and QR: MD (1:4). 

The procedure is described as follows: (i) a 2 mM silver nitrate (AgNO_3_) solution was prepared by mixing 0.0085 g of AgNO_3_ with 25 mL of distilled water. (ii) Similarly, a 2 mM quercetin solution was prepared by mixing 0.003 g of quercetin and the corresponding amount of microcrystalline cellulose or maltodextrin, depending on the system and the ratio tested according to Table 1, with 10 mL of a mixture of distilled water and ethyl alcohol (*v*/*v*). Initially, each quercetin solution had a pH = 4; therefore, the pH was adjusted to 7 with sodium hydroxide (NaOH). (iii) Finally, 2.5 mL of each quercetin solution system was added to the silver nitrate solution, left under stirring at 400 rpm for 2 h and left at rest for 24 h for the correct formation of the AgNPs, protected from sunlight. All the procedures were performed at room temperature.

#### 2.1.2. Antimicrobial Activity of AgNPs Synthesized at Different Ratios of Stabilizing Agent

The antimicrobial activity of AgNPs was tested by the disk diffusion method [7]. In this method, when bacteria do not grow around the disc and a clean ring is observed, this is considered as zone of inhibition. The AgNPs were tested against Gram-positive (*Staphylococcus aureus* ATCC-47077) and Gram-negative (*Escherichia coli* ATCC-25922). First, cellulose discs were impregnated with each AgNPs suspension for 24 h. On the other hand, the microbial strains of *E. coli* and *S. aureus* were inoculated in Trypticasein Soybean (TSB) nutrient broth and incubated at 35 ± 2 °C for 24 h. The turbidity of the culture broths was compared with 0.5 McFarland solutions, corresponding to 1–2 × 10^8^ colony forming units (CFU/mL). Using a sterile swab, the inoculated medium was spread uniformly on Mueller-Hinton agar plates as a solid growth medium. Once inoculated, the impregnated discs were placed on the growth medium and were left to incubate in an oven at 35 ± 2 °C for 24 h. As positive control, antibiotic discs of amoxicillin + clavulanic acid (AMC) were used. The concentration of the AMC discs was 30 µg in a 2:1 ratio, while the concentration of the AgNPs discs was 10 µg.

After, they were examined to verify the existence of a zone of inhibition, and this ring or halo was measured under a fume hood. The assays were performed in triplicate, and the mean values were calculated using ImageJ software.

#### 2.1.3. Characterization of AgNPs Synthesized at Different Ratios of Stabilizing Agent

The formation, size, shape, and long-term stability of nanoparticles prepared with pure quercetin and stabilizing agents were evaluated using the following characterization techniques:

(i) Change of color. When the quercetin solution is added to the silver nitrate solution, the color change of the solution from transparent to colloidal brown is immediately observed, which is characteristic of the silver nanoparticles; therefore, the observation of this color confirms the AgNPs formation. (ii) UV/Vis. Utilizing Ultraviolet-Visible spectroscopy (UV-Vis), the presence and stability of AgNPs were confirmed using a Lambda 35 model equipment from Perkin Elmer (Shelton, CT, USA), registering a wavelength from 200 to 800 nm. (iii) AFM. A Multimode 8-HR Nano Surfaces Atomic Force Microscope (AFM) from Bruker (Billerica, MA, USA), with high resolution was used in tapping mode on mica for their dispersion and morphology. The AFM uses a cantilever (or tip), varying the amplitude with which it presses the sample, allowing the reading of the topography and the size of our AgNPs. (iv) DLS. To verify the size of the AgNPs, a Delsa Nano C particle analyzer (Beckman Coulter, Brea, CA, USA) was used with the Dynamic Light Scattering (DLS) method, which sends a laser light beam through the sample solution and monitors the scattered photons at specific angles during short time intervals (microseconds) to determine the hydrodynamic size of the AgNPs. (v) Z-potential. The surface charge is the electrical potential between an aqueous solution and the stationary ion layer attached to the AgNPs. Such a measurement was obtained with a Delsa Nano C particle analyzer (Beckman Coulter, Brea, CA, USA), which gives information on the behavior of AgNPs in solution due to the distance between the particles, as well as the boundary separating them from the fluid and the thickness of the electrical double layer. (vi) *XRD analysis*. This analysis was performed in a D8 ADVANCE powder diffractometer with a Lynx Eye detector from Bruker (Billerica, MA, USA) and copper tube (wavelength 1.5406A), recording a range of 20–80° at 2θ angles.

#### 2.1.4. Minimum Inhibitory Concentration (MIC) and Minimum Bactericidal Concentration (MBC)

After the above characterizations, the systems with the best properties were subjected to the microdilution method in CLSI M7-A7 broth [12] to obtain their minimum inhibitory concentration (MIC) and minimum bactericidal concentration (MBC) against *S. aureus* and *E. coli*. First, a stock solution was prepared with the AgNPs and diluted with Mueller-Hinton broth to prepare ten different concentrations. An inoculum of each bacterium was prepared following the McFarland scale to reach a concentration of 0.5–2.5 × 10^8^ CFU/mL, diluted to 1:500 to obtain a final microorganism concentration of 0.5 × 10^3^. Next, the wells of a microplate were filled with 100 μL of each nanoparticle dilution per column, and then 100 μL of inoculum was added. A negative control was left to confirm the sterility of the medium, and a positive control containing the inoculum of each bacterium. Finally, they were incubated at 35 ± 2 °C with constant stirring for 24 h.

## 3. Results

Evaluation by color and UV/Vis of the AgNPs synthesized at different ratios of the stabilizing agent. The effect of the concentration of the stabilizing agent, both CM and MD, on the formation of AgNPs was systematically studied by preparing the solutions of AgNO_3_ and quercetin at the same concentration (2 mM and 1 mM, respectively) but with a different concentration ratio of the stabilizing agent to weight ratio of quercetin, as shown in Table 1. It is well-reported in literature [13] that AgNPs have a characteristic color of colloidal brown due to their characteristic surface plasmon resonance (SPR) properties. According to the results, after adding the quercetin solution, all the prepared systems change to amber brown color at different intensities. The corresponding images are in the inset of Figure 1. 

Spectroscopy UV/Vis is a simple technique that allows recording SPR spectra for colloidal nanoparticles. Their recordings indicate whether the nanoparticles are well dispersed, which happens when the absorption SPR has a relatively sharper peak; or if the nanoparticles have agglomerated or are no longer at the nanoscale, the SPR disappears [14]. UV-Vis analyses were carried out on all systems to evaluate the quality of each AgNPs solution. In the literature [6], it is reported that a SPR in a range of 400–450 nm corresponds to silver colloids. Figure 1 summarizes all the results, and as observed, most of them recorded a SPR at approximately 430 nm, which corresponds to the characteristic signal of AgNPs. To evaluate the effect of the stabilizing agent on the stability and long-term applicability of AgNPs, all the solutions were kept and evaluated at different times: 7, 14, 21, and 60 days. It is important to note that after 3 days, the system without any stabilizing agent, only pure quercetin, lost its colloidal appearance and the signal at 430 nm by UV-Vis; and at this day only agglomerates are visible to the naked eye. The above indicates that although it is possible to obtain AgNPs through this green route with pure quercetin and at pH 7, they are not stable and cannot be used for antimicrobial application.

Another information that we can obtain from the UV-Vis analyses is the effect of the stabilizing agent on the formation of AgNPs, since the same amount of AgNO_3_ and quercetin was used for all synthesis, but different ratios of the stabilizing agent were tested. When CM was used, only the systems QR:CM (1:1), QR:CM (1:2), and QR:CM (1:3) maintained the value of SPR at around 430 nm, but when more or less CM was used, a red shift in the absorption bad to longer wavelength was observed (from 430 nm to 448 for QR:CM (1:0.5) and to 444 nm for QR:CM (1:4). The above indicates that the particle size increased when the ratio of CM was less than or greater than the required for the stabilization. The same behavior was observed, only the ration QR:MD (1:1) was maintained the SPR at around 430 nm. In addition, when the ratio of stabilizing agent into the quercetin solution varied to (1:2) for both CM and MD, the absorption peak became narrower and the intensity increased, indicating an enhancement in the formation and amount of AgNPs [6].

On the other hand, it is observed that when a stabilizing agent was used (CM and MD), the AgNPs not only remained in the nano dimension but also increased their concentration, which was confirmed both visually and with the UV/Vis measurement, especially for the ratios of 1:1, 1:2, and 1:3. In addition, it was possible to observe that the ratio of the stabilizing agent is crucial since a low (1:0.5) or high (1:4) concentration does not maintain the nano dimension of the AgNPs, and they become agglomerated and lost. Finally, it was observed that the 1:2 ratio for both stabilizing agents presented the highest SPR, increasing the signal in a 92% and within the range of 430 and 440 nm.

Previous works have reported similar values for the size of AgNPs; for example, Othman et al. [15] reported a diameter range of their biosynthesized AgNPs between 10 and 34 nm through the mediation of fungal proteins from Aspergillus fumigatus but working in an alkaline medium. Meanwhile, Jain and Mehata [6] published that the average particle size obtained using Tulsi extract and quercetin as reducing agents was 14.6 and 11.35–18 nm, respectively. Finally, Carmona et al. [16] confirmed that using the extracts of leaves of the endemic hope Buddleja globosa, they were able to biosynthesize AgNPs with a mean diameter size of 16.37 ± 8.35 nm. Therefore, it is possible to establish that our AgNPs are well formed under neutral conditions and even at smaller sizes.

### 3.1. Antibacterial Activity of AgNPs Synthesized at Different Ratios of Stabilizing Agent

According to the UV/Vis results, it is possible to observe that the ratio and type of the stabilizing agent (CM or MD) significantly influence the formation of AgNPs, and to realize that both agents enhance the formation of AgNPs. To evaluate whether the different formation of AgNPs due to the different type and ratio of the stabilizing agents had an impact on the antimicrobial action of AgNPs, all the obtained systems were tested by the disc diffusion method against Gram-positive (*S. aureus)* and Gram-negative *E. coli* bacteria. The tests were carried out at a different time, day 1 and day 60, to also evaluate whether the AgNPs retained their activity for a long time. The results of the disc diffusion tests for all systems are shown in Table 2. It is observed that all systems show antimicrobial activity; it is important to mention that better performance was observed in the systems with enhanced formation of AgNPs, which were prepared with CM and MD at ratio 1:2. As discussed previously, AgNPs presented a different size and amount after the synthesis, depending on the type and ratio of a stabilizing agent, which agrees with the different inhibition zones found in this antimicrobial test. First, it is observed that the AgNPs system without the stabilizing agent QR (1:0) reduced its inhibition halo with the time, which was expected since the UV/Vis measurement indicated that the AgNPs became agglomerated, and the resonance plasmon was lost.

Regarding the systems with a stabilizing agent, it is possible to observe that only those that presented an increase in their resonance plasmon, QR:CM (1:2), QR:CM (1:3), QR:CM (1:4), QR:MD (1:1), QR:MD (1:2), and QR:MD (1:3), demonstrated better inhibition against both bacteria at day 60. This is attributed to the increased presence of AgNPs due to emulsifying agents’ presence and adequate ratio. In contrast, systems with a lower or higher ratio of emulsifying agents do not have stable nanoparticles for 60 days, which leads to a decrease in their antimicrobial capacity. It is worth mentioning that *S. aureus* presented wider halos due to this type of bacterium having a smooth single-layer cell wall, making it more susceptible to antibacterial agents, while *E. coli* has a double-layer and corrugated cell wall, which presents greater resistance [17] and consequently a smaller zone of inhibition when evaluated against AgNPs. The results in our previous work have demonstrated this same behavior, and AgNPs biosynthesized with quercetin have inhibited more Gram-positive bacteria. These results agree with research reported in the literature; for example, Chiu et al. [18] performed an investigation on the synthesis of AgNPs mediated by *Clinacanthus nutans*, they tested AgNPs against *S. aureus* and *E. coli* and found higher inhibition zones against *S. aureus* (11.35 mm and 11.52 mm) than against *E. coli* (9.22 mm and 9 mm). Similar results were obtained by Paul et al. [19], who synthesized silver nanoparticles using *Parkia roxburghii* leaf biomass and established that their AgNPs possess better inhibition against Gram-positive bacteria compared to Gram-negative bacteria. They attributed this to the absorption and accumulation of AgNPs in the cell wall of Gram-positive bacteria, which has a single thin membrane.

According to the results, the systems that demonstrated the best inhibition were those with a 1:2 ratio for both microcrystalline cellulose and maltodextrin. These results agree with the information provided in the UV/Vis analysis. This ratio (1:2) was the optimal proportion for the enhanced formation of AgNPs, because in these colloidal solutions there is a higher number of nanoparticles that will provide higher antibacterial activity.

Statistical analysis (Analysis of Variance ANOVA) was performed in the Minitab 18 software, followed by Tukey’s test to find significant differences due to the type and amount of stabilizing agent. According to the results, a high statistical significance level was found at *p* = 0.001 and *p* = 0.012 for *S. aureus*, and at *p* = 0.001 and *p* = 0.000 for *E. coli*. Therefore, the type and amount of stabilizing agent significantly affected the mean inhibition zone due to the difference in the values, with a significance level of *p* < 0.05.

Previously, Othman [15] reported an inhibition halo of 15 mm in diameter but only when working with a pH of 12, since those below this value did not present any inhibition. On the other hand, the work of Jain and Mehata [6] demonstrated an inhibition zone of 14 mm in diameter but only with some samples, while our AgNPs developed an inhibition halo of 8.30 mm but at pH 7.

### 3.2. Characterization of AgNPs by AFM, DLS, and Z-Potential of the Best AgNPs Systems

The above results are conclusive and demonstrate that to obtain AgNPs’ stable for 60 days, it is necessary to add an stabilizing agent in a ratio of 1:2. Therefore, it was decided to characterize the AgNPs prepared only with quercetin (QR (1:0)) and the best AgNPs systems (QR:CM (1:2) and QR:MD (1:2)) to evaluate their morphology, size, crystalline structure, and physicochemical properties.

AFM images were taken with values between 10 and 5 microns of the systems with the best ratio (1:2) of both emulsifying agent and the system containing only quercetin. In addition, the systems were also evaluated at different times (1 and 60 days) to observe their behavior, such as whether they presented agglomeration, an increase in size or changes in dispersion [2]. It was observed in the obtained images (Table 3) that in the systems prepared with the stabilizers, the dispersion of the AgNPs was not affected and they were even more dispersed and with a more defined morphology, practically spherical. Meanwhile, for the AgNPs system prepared with just quercetin (QR (1:0)), a very noticeable change was observed both in the dispersion and in the size of the AgNPs, since they formed large agglomerates, and neither the nanometer size nor the spherical shape was detected at day 60. Moreover, QR:CM (1:2) and QR:MD (1:2) systems were analyzed after 60 days, and it was noticed that the AgNPs with the smallest size were those prepared with microcrystalline cellulose, with a particle size around 50 nm, while with maltodextrin the particle size was between 50 and 70 nm.

In order to confirm the size, surface charge, and the effect due to the presence of the stabilizing agent, the hydrodynamic diameter and Z-potential of the AgNPs in the suspension were measured by Dynamic Light Scattering (DLS) at different times [15]. The results are summarized in Table 3, and it was observed that there is an increase in the size of the nanoparticles for all systems but especially for the one containing only quercetin. In this system, large agglomerates were now observed, and nanometer-sized particles are no longer detected. Meanwhile, for the systems prepared with maltodextrin and microcrystalline cellulose, there was also an increase, but the shape and size of the nanoparticles were maintained. It is possible to establish that the QR:CM (1:2) system was the one with values closer to its initial measurement, which confirms that this ratio and type of stabilizing agent helps to maintain the AgNPs at nanometer size, with values of 56.2 nm at Day 1 and 86.6 nm at day 2, according to DLS. In addition, the polydispersity index (PdI) of the nanoparticles was reported in Table 3. The results indicate that the presence of stabilizing agents in the colloidal AgNPs solution helps to avoid the agglomeration of the nanoparticles, as the QR (1:0) system (without any stabilizing agent) had the highest PDI index, which is due to the aggregation of the particles.

On the other hand, the data obtained employing Z-potential demonstrated that the best AgNPs system was the one in which microcrystalline cellulose was used, with values of −22.0 mV on day 1 and −34.6 mV on day 60. As observed, as time passed, its values increased above −30 mV, and it is known that the general dividing line between stable and unstable suspensions is generally taken at +30 or −30 mV. That is, particles with Z-potential values more positive than +30 mV or more negative than −30 mV are usually considered stable. Therefore, these values indicated that the AgNPs solution became more stable than at the beginning of the synthesis [2,13]. Meanwhile, the AgNPs prepared with just quercetin were the most unstable during the whole period evaluated. 

As observed, the Z-potential values for the AgNPs prepared with microcrystalline cellulose (QR:CM (1:2)) presents higher values than the one prepared with maltodextrin (QR:CM (1:2)). To understand this behavior, the Z-potential of microcrystalline cellulose and pure maltodextrin were measured at the same concentration and pH as in colloidal suspensions. The Z-potential of CM was −27.04 mV and that of maltodextrin was −23.58 mV, which means that the stability of maltodextrin in the solution is low compared to microcrystalline cellulose. The higher Z-potential value of CM may be attributed to it having a predominantly negatively charged surface [20]. In the literature, there are reported similar values of Z-potential of AgNPs, for example, Hebeish et al. [21] obtained AgNPs through a synthesis that uses garlic but with elevated operating conditions, a temperature of 70 °C, and pH 12, and reported the Z-potential value of −28 mV for their AgNPs; meanwhile, Ferreyra et al. [22], who used capers and microwaves, reported values of −41 ± 18 mV for their AgNPs. 

### 3.3. Evaluation of AgNPs by XRD

To confirm the crystalline structure of the AgNPs prepared without and with a stabilizing agent, XRD analysis was performed at day 1 and day 60. The results in Figure 2a) show peaks corresponding to the crystallographic planes (1 1 1), (2 0 0) and (2 1 1) related to 38°, 44° and 64° 2θ, respectively, for the AgNPs measured at day 1. Therefore, it is possible to establish that the use of stabilizing agents does not affect the crystal structure of AgNPs, since they had a face-centered cubic structure (FCC), according to the crystallographic planes detected [8]. Figure 2b) illustrates the AgNPs analyzed at day 60, and another interesting fact is that at day 60, the QR:CM (1:2) and QR:MD (1:2) systems maintain the characteristic peaks of this crystalline structure (FCC), as just a slight change in size was noted. With all this information, it can be affirmed that only the AgNPs obtained by a reduction with quercetin at neutral pH and that which are used stabilizing agents are stable and maintain their shape for 60 days; unlike the QR (1:0) AgNPs, which lose the definition of the peaks.

### 3.4. MIC and MBC of the Best AgNPs Systems

In addition to the antimicrobial susceptibility test by disk diffusion, MIC and MBC determination were carried out for the systems that obtained the best results and maintained their long-term stability: QR:CM (1:2) and QR:MD (1:2). Table 4 summarizes and illustrates the results obtained for both *S. aureus* and *E. coli* bacteria. The microdilution method was carried out for both AgNPs systems on day 1 and day 60, to also evaluate whether the AgNPs changed their long-term antimicrobial activity. However, the results demonstrated no significant differences with respect to the time at which AgNPs were evaluated, and the same MIC and MBC values were obtained; therefore, they are reported as a single value in Table 4. According to the observed for the system prepared with microcrystalline cellulose QR:CM (1:2), lower values for MIC and MBC for both *S. aureus* and *E. coli* are needed, and practically half the concentration is required compared to the system prepared with maltodextrin QR:MD (1:2). These results can be attributed to the fact that the size of the AgNPs obtained with CM is considerably smaller. The literature [3,23] has reported that the size and stability of the nanoparticles are fundamental to enhancing the antimicrobial activity.

### 3.5. Feasibility of Microcrystalline Cellulose and Maltodextrin as an Emulsifying Agent

The preparation of metallic nanoparticles depends not only on having a precursor agent that supplies the metallic element and the reducing agent that provides the electrons to reduce the metal ion to a zerovalent particle. A stabilizing agent is also needed to help keep the particles at a nanometer size [24]. These agents also provide colloidal stability and prevent agglomeration. The choice of a stabilizing agent is critical, as they define the morphology of the nanoparticle because it is adsorbed on its surface. Nowadays, it is desirable not to use aggressive chemical compounds, since they pollute and generate undesirable residues; therefore, green synthesis is an excellent alternative to transforming existing processes using natural materials and clean processes. The use of biopolymers such as natural polysaccharides allows the continued application of the green route in the formation of AgNPs, since it is a process that does not require additional energy or toxic tensoactives. Both microcrystalline cellulose and maltodextrin have hydroxyl groups that provide steric stability to the metal nanoparticles, thanks to ion-dipole interactions. Finally, the other advantage is that these polysaccharides have weak chemical interactions with the nanoparticles, which does not interfere negatively in the reaction and allows easy separation of the reaction mixture [25].

## 4. Conclusions

According to the results obtained by UV-Vis spectroscopy, AFM, DLS and XRD analysis, it can be stated that we are able to obtain AgNPs with a spherical form and nanometric size, with a crystalline structure and with good dispersion that avoids agglomeration, thanks to the stabilizing agents used. It is essential to highlight that the antimicrobial activity of the AgNPs biosynthesized at neutral pH and using emulsifying agents such as microcrystalline cellulose or maltodextrin was maintained in the long-term and was even better. On the contrary, at pH 7, the use of just quercetin does not maintain the AgNPs. The above was confirmed by UV/Vis spectroscopy, and the resonance plasmon at 430 nm disappeared, indicating that there are no nanoparticles in the aqueous solution. This was also observed with the naked eye; the colloidal brown color was no longer maintained, on the contrary, only agglomerated particles remained. In addition, it was notorious that when the solutions had the correct ratio of stabilizing agent, either microcrystalline cellulose or maltodextrin, the lifetime of the AgNPs was much longer (2 months). This can be explained thanks to both agents being composed of biomolecules that act as stabilizing agents. Thanks to these results, we can verify that the antimicrobial activity of the AgNPs was not lost and even increased. According to the results, it can be corroborated that this green synthesis for the obtaining of long-term stable AgNPs is viable. This route does not work with corrosive or toxic reagents or complicated operations such as high temperature, vacuum, or alkaline conditions, but rather uses natural compounds such as quercetin, microcrystalline cellulose, and maltodextrin at neutral pH. This method has the advantage of being simple and fast, in addition to being able to have stable AgNPs for the long-term. The tests allow us to find the best ratio of reducing agent:stabilizing agent (1:2) for microcrystalline cellulose and maltodextrin. These excellent characteristics provide them with antimicrobial activity and stability up to 2 months under normal storage conditions. All of the above is due to the presence of stabilizing agents used in the food industry.

## Figures and Tables

**Figure 1 nanomaterials-12-03545-f001:**
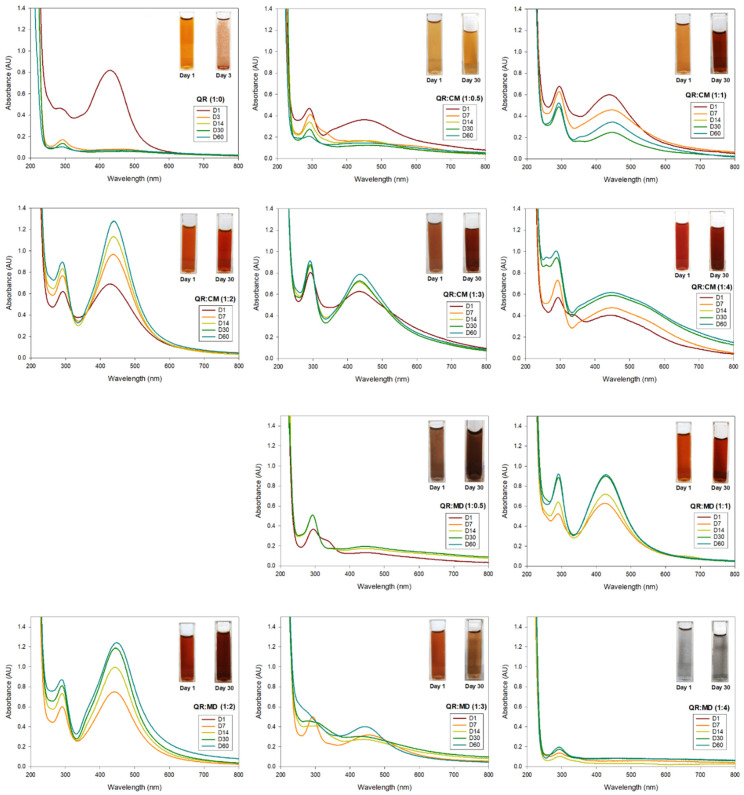
UV-VIS absorption spectra and color change for each AgNPs system at different ratios of quercetin and stabilizing agent: microcrystalline cellulose and maltodextrin for different times.

**Figure 2 nanomaterials-12-03545-f002:**
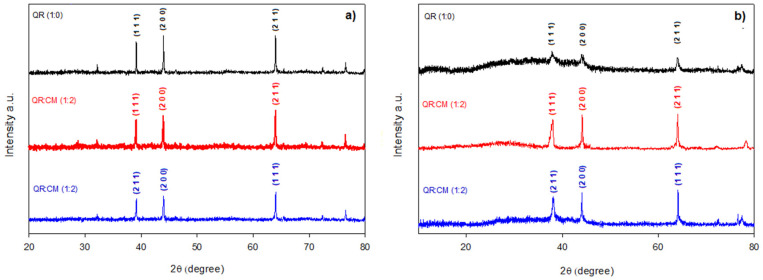
XRD patterns for the best AgNPs systems at different times: (**a**) Day 1 and (**b**) Day 60.

**Table 1 nanomaterials-12-03545-t001:** Preparation of AgNPs at different ratios of stabilizing agent.

AgNPs System	AgNO_3_	Quercetin	Stabilizing Agent
	(g)	(g)	(g)
QR (1:0)	0.0085	0.003	0
			** *Microcrystalline cellulose* **
QR:CM (1:0.5)	0.0085	0.003	0.0015
QR:CM (1:1)	0.0085	0.003	0.003
QR:CM (1:2)	0.0085	0.003	0.006
QR:CM (1:3)	0.0085	0.003	0.009
QR:CM (1:4)	0.0085	0.003	0.012
			** *Maltodextrin* **
QR:MD (1:0.5)	0.0085	0.003	0.0015
QR:MD (1:1)	0.0085	0.003	0.003
QR:MD (1:2)	0.0085	0.003	0.006
QR:MD (1:3)	0.0085	0.003	0.009
QR:MD (1:4)	0.0085	0.003	0.012

**Table 2 nanomaterials-12-03545-t002:** Antibacterial activity by disc diffusion method of all AgNPs systems.

AgNPs System	Mean Inhibition Zone (mm)			
	*S. aureus*		*E. coli*	
	Day 1	Day 60	Day 1	Day 60
QR (1:0)	10.07 ^±0.77^	9.87 ^±0.73^	7.33 ^±0.51^	7.14 ^±0.38^
QR:CM (1:0.5)	9.35 ^±0.54^	8.95 ^±0.73^	7.02 ^±0.52^	6.62 ^±0.18^
QR:CM (1:1)	9.19 ^±0.50^	9.89 ^±0.45^	7.09 ^±0.36^	7.24 ^±0.29^
QR:CM (1:2)	10.16 ^±0.74^	11.37 ^±0.44^	7.89 ^±0.62^	8.54 ^±0.45^
QR:CM (1:3)	9.37 ^±0.78^	10.82 ^±0.22^	7.15 ^±0.32^	7.42 ^±0.26^
QR:CM (1:4)	10.18 ^±0.50^	9.89 ^±0.63^	7.30 ^±0.35^	7.25 ^±0.16^
QR:MD (1:0.5)	-	-	-	-
QR:MD (1:1)	9.55 ^±0.63^	10.17 ^±0.78^	7.48 ^±0.69^	7.68 ^±0.52^
QR:MD (1:2)	10.11 ^±0.38^	11.23 ^±085^	7.90 ^±0.30^	8.18 ^±0.25^
QR:MD (1:3)	9.97 ^±0.81^	10.10 ^±0.87^	7.01 ^±0.22^	7.87 ^±0.36^
QR:MD (1:4)	-	-	-	-
Positive control (AMC)	15.28 ^±0.29^	15.34 ^±0.18^	10.43 ^±0.40^	10.30 ^±0.14^

**Table 3 nanomaterials-12-03545-t003:** AFM images, hydrodynamic diameter, PdI, and Z potential for the best AgNPs systems at different times.

System	AFM		Hydrodynamic Diameter (nm)		Polydispersity Index (PDI)		Z-Potential (mV)	
	Day 1	Day 60	Day 1	Day 60	Day 1	Day 60	Day 1	Day 60
QR (1:0)	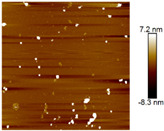	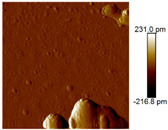	79.1 ^±4.4^	14741.5 ^±370.1^	0.18 ^±0.04^	0.47 ^±0.05^	−0.08 ^±6.3^	−13.3 ^±4.8^
QR:CM (1:2)	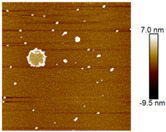	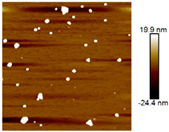	56.2 ^±1.1^	86.6 ^±3.4^	0.16 ^±0.02^	0.18 ^±0.03^	−22.0 ^±3.2^	−34.6 ^±4.3^
QR:MD (1:2)	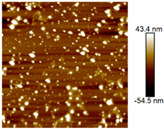	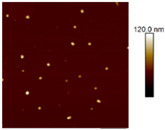	77.7 ^±3.9^	321.6 ^±16.0^	0.17 ^±0.02^	0.20 ^±0.04^	−14.8 ^±4.9^	−22.0 ^±5.5^

**Table 4 nanomaterials-12-03545-t004:** MIC and MBC for the best AgNPs systems.

System	*S. aureus*		*E. coli*	
	MIC	MBC	MIC	MBC
QR:CM (1:2)	62.5 µg/mL	250 µg/mL	125 µg/mL	500 µg/mL
QR:MD (1:2)	125 µg/mL	500 µg/mL	250 µg/mL	1 mg/mL

## Data Availability

Not associated data.

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
