# Peer review of "Green Route to Produce Silver Nanoparticles Using the Bioactive Flavonoid Quercetin as a Reducing Agent and Food Anti-Caking Agents as Stabilizers"

_nanomaterials, 2022, doi:10.3390/nano12193545_

Round 1

Reviewer 1 Report

The research presented in the manuscript is in the scope of the journal readership and could be of interest but currently is written in a very general way and without the real focus to follow the research suggested by the title: Green Route to Produce Silver Nanoparticles Using the Bioactive Flavonoid Quercetin as a reducing agent and Food Anti-3 caking Agents as Stabilizers.

In the introduction is mentioned about Microcrystalline cellulose (CM) and maltodextrin (MD), which " were used at different weight ratios regarding quercetin to find the optimal ratio to enhance synthesis, long-term stabilization, and antimicrobial properties against Gram-positive and Gram-negative bacteria" and there is no mention about the AgNPs formation....

In the section about AgNPs synthesis, the important parameters, such amounts, volumes or concentrations, temperature, are not provided.

In the lines 218-219, the written sentence needs to be verified by checking information from Ref. 15 and clarified if the antimicrobial agents mentioned there are AgNPs: " while E. Coli, having a double-layer and corrugated cell wall, presents greater resistance and consequently a smaller zone of inhibition [15]."

In Table 1, where is the information about disc diffusion method and why units are not provided?; where there are positive controls with antibiotics and silver ions?...; there are no significant differences in the results...

In Table 2, it would be good to provide the size and zeta potential monitoring during the period of 60 days to see the trend of gradual size ot zeta potential increase or maybe the stabilization of size after certain day...

In the lines 272-274, for the sentence: " These results can be attributed to the fact that the size of the AgNPs obtained with CM is considerably smaller. The literature has reported that the size and stability of the nanoparticles are fundamental to enhancing antimicrobial activity." Reference should be provided.

For Table 3, the MIC/MBC should be repeated and done in various days between day 1 and 60 to see the size influence on the bio-potency.

The finding from this green synthesis should be compared to other known in the literature bio-synthesized AgNPs and their physico-chemical characteristics, as well as antibacterial activity.

Author Response

Dear Reviewer,

Please find attached our response to your requirements.

Reviewer 2 Report

Dear Authors,

I read the paper Green Route to Produce Silver Nanoparticles Using the Bioactive Flavonoid Quercetin as a reducing agent and Food Anti-caking Agents as Stabilizers (ID: nanomaterials-1896236) and I concluded that the manuscript cannot be published in Nanomaterials.

Major observations are as following:

1. The presentation style in English must be improved (see, for example, the purpose of the paper, lines 89-92, which has too long sentences; or page 3, line 113 - .. which the ring around the disk in which the bacteria ... – repetitions should be avoided).

There are some confusing expressions (as an example, first and third phrases from Abstract need to be reformulated).

- Page 2, line 57 – .... plants, roots and fruits .. - it needs to be reformulated (maybe, the roots and fruits of plants?)

- Page 8, lines 297 – 301 – the phrase is very long and difficult to understand. The same observaion for the lines 306 – 315.

The above are just a few mentions. Such confusions are encountered throughout the article.

2. Abstract – line 25 - „...and microcrystalline cellulose and maltodextrin as agents...”.  The type of agent must be mentioned. Are they stabilizing agents, surfactants?

3. Introduction

- Page 2, line 62 – Do references [5] and [6] contain comparison between plant extract and reagent-grade quercetin? I have not seen such comparison in these articles.

4. Materials and Methods

- Page 3 - Synthesis of silver nanoparticles (AgNPs) at a different ratio of stabilizing agent - What were the pH solutions before adjustment with sodium hydroxide?

- Page 3, line 108 - point (iii) corresponds to the AgNps synthesis from the system with pure quercetin QR (1:0)?

5. Results

- Page 4 – Reference [14] is not about silver colloids as the authors mentioned.

- Page 5, Figure 1 – the resolution must be improved.

- Page 5, lines 206, 207 -... and again the influence of AgNPs formation depends on the type and ratio of a stabilizing agent ... - it does not make sense because the nanoparticles are already obtained and the same particles are used for characterization and test.

- Is there any explanation why the samples with 1:2 ratio ((QR:CM / QR:MD) showed the best inhibition?

- Page 6, lines 238, 239 - .. and neither nanometric nor spherical sizes were detected... - Appropriate expression is „... nanometric size nor spherical shape ...”.

- Page 6, lines 241, 242 – „.. with measurements between 50 nm ...”. - Appropriate expression is „with particle size between 50 and (what value?) nm”

- Page 7 – Z-potential values - The system with microcrystalline cellulose is better than the one with maltodextrin. What is the explanation?

- DLS – for the average hydrodynamic size is necessary to add +/- error. The polydispersity index (PDI) should also be specified. Characterizations such as XRD, FTIR and TEM are also missing.

- Page 8, lines 279, 280 – „.. and the reducing agent that provides the electro to reduce the metal ion to a zerovalent particle ...„. - it can not be understood, maybe correct is „electrons”?

6. The authors have already obtained Ag particles (references 8 and 9) at pH 7 using quercetin as a reducing agent, and the antimicrobial activity was tested against S. aureus and E. coli, as in the current article. What is different in this article for the sample obtained from the QR 1:0) system? A comparison of the Ag nanoparticles obtained in this study with those obtained previously would have been necessary. For example, it is possible to synthesize AgNPs with long-term stability and their effect remained after 180 days, as it results from reference [8]. In this case, Ag samples obtained from the systems with QR:CM and QR:MD have better properties?

Moreover, the authors should have compared the properties of the synthesized silver nanoparticles with those for silver nanoparticles obtained by other (green) methods using quercetin. There are articles in the literature on this subject.

Additional observation: For References section see Instructions for Authors.

Author Response

(The authors gave the same response as above.)

Round 2

Reviewer 1 Report

- With the new sentence: ” It is worth mentioning that S. aureus presented wider halos due to this type of bacterium has a smooth single-layer cell wall making it more susceptible to antibacterial agents, while E. coli has a double-layer and corrugated cell wall, which presents greater resistance [17] and consequently a smaller zone of inhibition when evaluated against AgNPs. there is still problem as the authors don’t present example of other AgNPs being more active against Gram-positive bacteria than against Gram-negative bacteria; usually silver is more potent against Gram-negative bacteria as is able to break the outer membrane built of lipopolysaccharides but has difficulties in crossing the thick peptidoglycan cell wall...As authors state something different, then should support it by the reference....In their previous work with just quercetin and silver, the complex was also more potent against Gram-positive bacteria? - In Table 2, the authors have added the information about positive control to be antibiotic discs of amoxicillin + clavulanic acid (AMC) but the authors should provide the details about the concentrations of these both components in AMC and the final volume used; the same in case off all sample with AgNPs (the used amount/volume; in micromolar values or mg/mL) for this disk diffusion experiments.

Author Response

Dear Reviewer,

We have attended your comments as follow:

The concentration of the AMC discs was 30 µg in a 2:1 ratio, while the concentration of the AgNPs discs was 10 µg.

The results in our previous work have shown this same behavior, AgNPs biosynthesized with quercetin, have inhibited more Gram-positive bacteria. These results agree with research reported in the literature; for example, Chiu et al. [18] performed an investigation on the synthesis of AgNPs mediated by Clinacanthus nutans, they tested AgNPs against S. aureus, and E. coli and found higher inhibition zones against S aureus (11.35 mm and 11.52 mm) than against E. coli (9.22 mm and 9 mm). Similar results were obtained by Paul et al [19], who synthesized silver nanoparticles using Parkia roxburghii leaf biomass and established that their AgNPs possess better inhibition against Gram-positive bacteria compared to Gram-negative bacteria. They attributed this to the absorption and accumulation of AgNPs in the cell wall of Gram-positive bacteria, which has a single thin membrane.

This information is already included in the manuscript.

Thank you

Reviewer 2 Report

Dear Authors,

The manuscript has been improved. I have only some minor observations:

1. - lines 31and 32 - repetition should be avoided (.... another stabilizing agent used ... as stabilizing agents). 

As a recommendation: ... microcrystalline cellulose and maltodextrin as stabilizing agents ...

or another wording that the authors consider appropriate.

2. Figs. 2a and 2b - add the unit (degree) for OX axe.

Peaks should be indexed as (111), etc, not as degrees.

Line 354 - crystallographic plane is (111), not (1111).

I agree with the publication of this paper after minor revision.

Author Response

Dear Reviewer,

We have modified the paper based on the last comments. Please find attached the revised document with the few modifications.

Sincerely,
